# Uncertainty budgets of major ozone absorption cross-sections used in UV remote sensing applications

Mark Weber<sup>1</sup>, Victor Gorshelev<sup>1</sup>, and Anna Serdyuchenko<sup>1\*</sup>

<sup>1</sup>Institut für Umweltphysik, Universität Bremen FB1, PO Box 330 440, D-28334 Bremen, Germany
\*now at: OHB System AG, Manfred-Fuchs-Straße 1, D-82234 Weßling, Germany

Correspondence to: Mark Weber (weber@uni-bremen.de)

Abstract. Detailed uncertainty budgets of three major UV ozone absorption cross-section datasets that are used in remote sensing application are provided and discussed. The datasets are Bass-Paur (BP), Brion-Daumont-Malicet (BDM), and the more recent Serdyuchenko-Gorshelev (SG). For most remote sensing application the temperature dependence of the Huggins

- ozone band is described by a quadratic polynomial in temperature (Bass-Paur parameterisation) by applying a regression to the cross-section data measured at selected atmospherically relevant temperatures. For traceability of atmospheric ozone measurements uncertainties from the laboratory measurements as well as from the temperature parameterisation of the ozone cross-section data are needed as an input to detailed uncertainty calculation of atmospheric ozone measurements. In this paper the uncertainty budgets of the three major ozone cross-section datasets are summarised from the original literature.
- The quadratic temperature dependence of the cross-section datasets is investigated. Combined uncertainty budgets is provided for all data sets based upon Monte Carlo simulation that includes uncertainties from the laboratory measurements as well as uncertainties from the temperature parameterisation. Between 300 and 330 nm both BDM and SG have an overall uncertainty of 1.5%, while BP has a somewhat larger uncertainty of 2.1%. At temperatures below about 215 K, uncertainties in the BDM data increase more strongly than the others due to the lack of very low temperature laboratory measurements (lowest temperature of BDM available is 218 K).

# 1 Introduction

The three ozone absorption cross-sections in common use for many remote sensing applications are the Bass Paur (BP) data (Bass and Paur 1985; Paur and Bass, 1985), the Daumont-Brion-Malicet (BDM) data (Daumont et al., 1992; Brion et al., 1993; Malicet et al., 1995) and the very recent Serdyuchenko-Gorshelev (SG) data (Gorshelev et al., 2014; Serdyuchenko et

al., 2011; 2014). While the data from BDM and SG are absolute cross-section measurements, the BP data were scaled to the so-called Hearn value at the Hg line wavelength (253.65 nm). The standard retrievals applied to the ground Brewer and Dobson spectrophotometer data use the BP data (e.g. Redondas et al., 2014), while the satellite community uses any of the three data sets or other data (WMO-GAW, 2015; ACSO, 2010; Orphal et al., Absorption cross-sections of ozone - Status report 2015, manuscript in preparation).

- For the review of uncertainties, original publications reporting on results of the experimental work were considered first. Since BP data was absolutely scaled using Hearn data (Hearn, 1961), the latter is also included in this review. There is a lack of consistency in the presentation of measurement uncertainty budgets across different papers. Neither of the publications uses the guidelines and recommended definitions as outlined in JCGM-100 (2008). An attempt to analyze and harmonize the reported uncertainties is made in the following sections. In some cases not all measured quantities were reported and in most cases detailed description of the data processing procedures is missing. It is very likely that the published measurement
- uncertainties are incomplete and the overall uncertainties thus underestimated.

For gaseous species the Beer-Lambert law can be written as:

In this paper we start with a brief summary on measurement principles in the laboratory (Section 2), followed by a review of the uncertainties of the UV ozone cross-section data (Section 3). In Section 4 the temperature dependence in the three major ozone cross-section datasets are discussed followed by Section 5, which summarises the Monte-Carlo simulation to

40 obtain the overall uncertainty budgets of the major datasets. Section 6 provides a summary and conclusion.

## 2 Measurement technique

Ozone absorption cross-sections are produced by performing spectroscopic measurements and subsequent analysis to convert the recorded spectra into absorption cross-sections in units of  $cm^2/molecule$ . The absorption spectroscopy is based on the Beer-Lambert law, which describes the attenuation of the light intensity transmitted through the absorbing medium.

$$I(\lambda) = I_0(\lambda) \int_0^L e^{-n(T,p,\ell) \cdot \sigma(\lambda,T)} d\ell.$$
(1)

Here  $I_0(\lambda)$  is the light intensity in the absence of absorbing molecules (background), n is the absorbing gas number density, which is generally a function of temperature T, pressure p, and the position  $\ell$  along the beam path, L is the total absorption path length and  $\sigma$  [*cm*<sup>2</sup>/*molecule*] is the wavelength-dependent (and normally also temperature-dependent) absorption cross-section.

In a laboratory environment it is possible to control the experimental conditions with sufficient precision, so that the number density  $n [cm^{-3}]$  is assumed to be homogeneously distributed along the absorption path of a known length L [cm]. From measurements of other parameters, such as T [K] and p [Pa], the value of n is calculated assuming the ideal gas law ( $p = n \cdot k_B \cdot T$ ,  $k_B$  Boltzmann constant). In this case Eq. 1 can be transformed to:

$$A(\lambda, T) = 1 - I/I_0 = 1 - \exp\left[-\sigma(\lambda, T) \cdot \mathbf{n} \cdot \mathbf{L}\right],$$
(2)

where A is the unitless absorbance. The unitless optical density (OD) is then expressed as

$$OD(\lambda, T) = \ln (I/I_0) = \sigma(\lambda, T) \cdot n \cdot L .$$
(3)

- Absolute cross-sections can be derived from the optical density if species concentration, temperature and absorption path length are known. Since ozone is a reactive and highly explosive gas, many measurements were done using a flow of oxygen/ozone mixture, where the partial pressure of ozone is unknown and values from other published absolute ozone cross-section measurements are used to find a scaling factor that converts the measured optical density into absorption crosssection (e.g. Chehade et al., 2013a, b)
- Alternatively, measurements performed at selected wavelengths with a special attention to control of the experimental parameters are used to calibrate the relative cross-sections (optical densities). The latter was done for the BP data (Bass and Paur 1985, Paur and Bass, 1985) as all their relative spectra were scaled to the ozone absorption cross-section value at 253.65 nm (mercury line) as reported by Hearn (1961). Thus the uncertainties in the reference data propagate into the calibrated spectra.
- Depending on the kind of spectrometer used for broadband (Fourier transform, grating) or single wavelength measurements, registered spectra are inevitably subject to multiple *sources of uncertainties* - stochastic intensity variations caused by detector noise, light source intensity fluctuations etc. (JCGM-100, 2008). Spectral *random error* can be characterized by the signal-to-noise ratio (*SNR*), and one of the ways to improve the quality of the measurements is acquisition of multiple spectra obtained under *repeatable conditions* (JCGM-100, 2008). Uncertainty of the resulting average
- value is represented by the standard deviation of the mean.

Spectrometers are characterized by the spectral resolution and wavelength calibration to some reference values, which influences the wavelength uncertainty of the produced data. For example, dispersion-based instruments can be wavelength-calibrated using isolated atomic emission lines of Hg or Cd lamps, and Fourier-transform spectrometers are auto-calibrated with the built-in stabilized He-Ne laser.

- Instrumental uncertainties of other measured quantities temperature T, ozone (partial) pressure *p*, absorption path length L are also contributing to the total absorption cross-section uncertainty. For broadband laboratory measurements covering a large wavelength range, like the Hartley-Huggins band of ozone, cross-section values change by up to seven orders of magnitude, so that optical density spectra are recorded using different combination of cell lengths and partial pressures (e.g. Gorshelev et al. 2014, Serdyuchenko et al. 2014). The various spectra are then concatenated to cover the entire spectral
- range, which leads to additional uncertainties.

# 3 Review of reported uncertainties

#### 3.1 Uncertainty budget of Hearn

Dating back to 1961, Hearn reported on ozone absorption cross-sections at six selected wavelengths, of which the value at  $\lambda = 253.65$  nm is of particular interest, since it has been measured in many studies and is considered a standard reference (see Viallon et al. (2015) and references therein). The following information of the laboratory measurements by Hearn (1961)

are known:

- spectral resolution: 0.09 nm (Hg emission line width at 253.65 nm)
- temperature: 295 K
- temperature uncertainty: not reported ("Errors due to the variation of the temperature of the ozone-oxygen mixture
  - during the experiment are quite negligible. The apparatus was housed in a cellar, the temperature of which was thermostatically controlled.")
- Absolute scaling: pressure observation of the pure O3 -> O2 decomposition

Table 1 provides the original notation of uncertainty budget from Hearn (1961). The uncertainties are divided into type A and type B uncertainties (JCGM-100, 2008). Type A uncertainty means that it is derived from a statistical analysis and an abaarned frequency distribution. Type B uncertainties are not derived from a statistical analysis and it assumes a probability.

observed frequency distribution. Type B uncertainties are not derived from a statistical analysis and it assumes a probability distribution that is based upon past experiences or is derived from external specifications. The breakdown of evaluation type of the uncertainties is color coded in Table 1.

Table 1: Summary of uncertainties as reported by Hearn (1961). Light gray highlighting stands for type A, dark gray for type B uncertainties.

| "Random errors"                                 |                                                     |                   |          |          |
|-------------------------------------------------|-----------------------------------------------------|-------------------|----------|----------|
| Wavelength uncertainty at                       | RMS deviation                                       | Absorption length | Pressure | Total SD |
| 253.65 nm                                       | (mean of 6 observations)                            | (0.744 cm)        |          | (RMS)    |
| 0.09 nm                                         | 1.05 %                                              | 0.54 %            | 0.81 %   | 1.4 %    |
| Type B                                          | Type A Type B                                       |                   |          |          |
| "Systematic errors"                             |                                                     |                   |          |          |
| Wavelength uncertainty                          | Correction for stray light Correction for companion |                   | anion    |          |
| at 253.65 nm                                    |                                                     |                   |          |          |
| 0.09 nm                                         | 0.0 -                                               |                   |          |          |
| "Best estimates of the absorption coefficients" |                                                     |                   |          |          |
| Wavelength uncertainty                          | Molecular absorption cross-section                  |                   |          |          |
| at 253.65 nm                                    |                                                     |                   |          |          |
| 0.09 nm                                         | $114.7 \pm 2.4 \ 10\text{-}19 \ \mathrm{cm}^2$      |                   |          |          |

Little to no detail is provided on the accuracy of the instruments used during measurements. Given the reported  $\pm 2.4 \times 10^{-19}$  cm<sup>2</sup> interval around the 114.7x10<sup>-19</sup> cm<sup>2</sup> value of absorption cross-section and assuming rectangular distribution of possible values (JCGM-100, 2008), the relative standard measurement uncertainty of the ozone absorption cross-section is estimated at 1.4%. Adding up all uncertainties the Hearn value has a precision of 2%. It should be noted here that more recent measurements (Viallon et al., 2015, Jansson C. et al., Absolute ozone absorption cross section at 253.65 nm revisited, manuscript in preparation) indicate lower values for the mercury line that lies about 1.4 to 1.8% lower than Hearn's value, but it is within the uncertainty of the Hearn's experiment. Compared to all available measurements reported, Hearn's value is close to the upper range of values (e.g. Sofen et al., 2015, WMO-GAW, 2015),

# 3.2 Uncertainty budget of BP

- The team of Bass and Paur (Bass and Paur 1985, Paur and Bass, 1985) provided cross-section data for a broad spectral range and at several temperatures. The following information is available from the BP data:
  - Wavelength range: 245 343 nm
  - Spectral resolution: 0.025 nm
  - Wavelength grid: 0.05 nm
- Wavelength calibration: 23 points between 200 and 365 nm (Hg, Cd, Zn lines)
  - Temperatures: 203 K, 218 K, 228 K, 243 K, 273 K, 298 K

Table 2 summarises the uncertainties of BP data, with a type A/B breakdown color-coded in different gray shadings. The relative standard measurement uncertainty of the BP ozone absorption cross-section is stated to be around  $\pm 1\%$ . It seems to be an underestimation, since BP relative spectra were scaled to the Hearn value at 253.65 nm, which is reported with a 1.4% relative standard measurement uncertainty (see Section 3.1).

Table 2: Absolute uncertainties reported by Bass and Paur (1985) and Paur and Bass (1985). Light gray highlighting stands for type A , dark gray for type B uncertainties.

| Uncertainty                      | Values                                      |
|----------------------------------|---------------------------------------------|
| Wavelength uncertainty           | 0.025 nm                                    |
| Uncertainty in the transmittance | 2 in 105 (arising from counting statistics) |
| determination                    |                                             |
| Sample temperature stability     | better than 1 K                             |
| Temperature uncertainty          | 0.25 K                                      |
| Pressure measurement uncertainty | 1 mbar                                      |
| Absolute scaling                 | using the value of Hearn at 253.65 nm       |

# 3.3 Uncertainty budget of BDM

Ozone absorption cross-sections provided by Brion et al. (1993), Daumont et al. (1992) and Malicet et al. (1995) further extend the wavelength coverage (into the visible) compared to BP. The following information is available on the experimental details for BDM data in the Hartley-Huggins ozone absorption band:

- Wavelength range: 195 345 nm (except at 273 K: 300–345 nm)
  - Spectral resolution: 0.01 nm
  - Wavelength grid: in steps of 0.01 nm
  - Concatenation: 15 nm wide spectral cuts, 5 nm overlap
  - Number of spectra averaged: 10
- Temperatures: 218 K, 228 K, 243 K, 273 K, 295 K
  - Temperature uncertainty: from 0.05 K @ 295 K to 0.3 K @ 218 K

- Light source reference spectra: recorded before and after the ozone spectra

- Absolute scaling: measurements of total pressure

Table 3 provides the original notation of the uncertainty budget of BDM data with Type A / Type B breakdown colorcoded in gray shadings. The information on the relative standard measurement uncertainty of the BDM ozone absorption cross-section is wavelength-dependent (see last row of Table 3).

 Table 3: Summary of uncertainties as reported by Brion et al. (1993), Daumont et al. (1992) and Malicet et al. (1995). Light gray highlighting stands for type A, dark gray for type B uncertainties.

| Quantity               | Uncertainty                             |
|------------------------|-----------------------------------------|
| Optical density        | 1 % (for $\lambda$ < 335 nm)            |
| Optical path           | 0.05 %                                  |
| Ozone pressure         | 0.1 %                                   |
| Impurities             | < 0.1 %                                 |
| Temperature            | from 0.02 % @ 295 K                     |
|                        | up to 0.15 % @ 218 K                    |
| Wavelength             | < 0.05 % (Hartley band 200 – 280 nm)    |
|                        | < 0.8 % (Huggins band 280 – 340 nm)     |
| Total systematic error | 1.3 – 1.5 % (Hartley band 200 – 280 nm) |
|                        | 1.3 – 3.5 % (Huggins band 280 – 340 nm) |
| Random error RMS       | $0.3 - 2.2$ % (for $\lambda < 340$ nm)  |

# 3.4 Uncertainty budget of SG

Ozone absorption cross-sections reported by Serdyuchenko et al. (2014) were obtained for 11 temperatures in a wide spectral range using two spectrometers (Fourier-transform and Echelle-grating spectrometers). Tables 4 and 5 summarize the information on the experimental details and uncertainties for the SG cross-section data.

In the 213-350 nm wavelength region the relative standard measurement uncertainty of the SG ozone absorption crosssection is wavelength-dependent and ranges between 1 - 3 %. The dominating uncertainty source is the statistical repeatability of the spectral registration, influenced by stability of the light source and detector noise. These two factors have a greater impact when the intensities of the spectra contributing to the OD calculation either differ greatly (strong absorption,

close to saturation) or are very close to each other. This effect is demonstrated in Figure 1, showing the concatenated OD spectrum and relative uncertainties of the corresponding constituent spectral cuts. The latter were calculated according to law of propagation of uncertainty using standard deviation (variances) and mean values of I and  $I_0$  spectra which determine the optical density OD (see Eq. 3).

Table 6 summarises the uncertainties for all cross-sections datasets discussed here.

<sup>145</sup> 

Table 4. Experimental details and statistical uncertainty of OD spectra for different wavelength regions for SG data (Serdyuchenko et al., 2014).

| Region,<br>[nm] | Spectrometer,<br>detector | Resolution<br>[nm] | Calibration | Path,<br>[cm] | Lamp<br>stability*<br>[%] | Optical<br>density |
|-----------------|---------------------------|--------------------|-------------|---------------|---------------------------|--------------------|
| 213 - 310       | Echelle, ICCD             | 0.018              | Relative    | 5             | D <sub>2</sub> , 0.5      | 0.5 - 2            |
| 310 - 335       | FTS, GaP                  | 0.01               | Absolute    | 135           | Xe, 2                     | 0.1 - 2            |
| 335 - 350       | FTS, GaP                  | 0.012              | Relative    | 270           | Xe, 1                     | 0.1 - 1            |
| 350 - 450       | Echelle, ICCD             | 0.02               | Relative    | ~2000         | Xe, 1                     | 0.05 - 1           |
| 450 - 780       | FTS, Si                   | 0.02-0.06          | Absolute    | 270           | W, 0.2                    | 0.05 - 2           |
| 780 -1100       | FTS, Si                   | 0.12-0.24          | Relative    | 270           | W, 0.2                    | 0.001 - 0.1        |

\* during the entire measurement

D2, Xe - deuterium and xenon discharge lamps, W - tungsten filament lamp

# 170

Table 5. Summary of absolute and relative measurement uncertainties for SG dataset. Light gray highlighting stands for type A, dark gray for type B uncertainties.

| Systematic uncertainty (abs)                                  | (rel) [%]   | Statistical uncertainty (abs)               | (rel) [%] |  |
|---------------------------------------------------------------|-------------|---------------------------------------------|-----------|--|
| Ozone impurity:                                               | 0.005       | Ozone initial pressure                      | 

#### Table 6. Uncertainties for Hearn, BP, BDM and SG ozone cross-sections.

| Dataset           | Scaling method       | Type A<br>(Statistical) | Туре В              | Relative standard<br>measurement<br>uncertainty, [%] |
|-------------------|----------------------|-------------------------|---------------------|------------------------------------------------------|
| Hearn (253.65 nm) | Absolute, pure ozone | 1.05                    | _                   | 1.4                                                  |
| BP                | Using Hearn          | 1                       | 2.1                 | > 2.1                                                |
| BDM               | Absolute, pure ozone | 0.9 - 2.2               | 1.3 (Hartley)       | 2 - 3                                                |
|                   | -                    |                         | 1.3 - 3.5 (Huggins) | 2 - 4                                                |
| SG                | Absolute, pure ozone | 1 - 2.2                 | 0.4 - 1.7           | 1.1 - 3                                              |