# Peer review of "Uncertainty budgets of major ozone absorption cross-sections used in UV remote sensing applications"

_Atmospheric Measurement Techniques, 2016_

## Referee Comment (RC1) · Anonymous Referee #1 · 18 May 2016

**Review of the submitted article:**

**Uncertainty budgets of major ozone absorption cross-sections used in UV remote sensing applications**

Mark Weber, Victor Gorshelev, and Anna Serdyuchenko

General comments:

The submitted article presents a critical review of three published datasets of ozone absorption cross-sections in the Huggins band, including data previously published by the same authors, with a focus on their uncertainty. In that regard it addresses an issue which has often been underestimated. It should bring valuable outputs for the community of scientists involved in ozone monitoring with remote sensing instruments, in providing a sound base to select the most appropriate dataset to finally improve the confidence in measurements. To calculate uncertainties associated with cross-sections at any temperature, the authors have chosen to use Monte-Carlo simulations. This tool is recommended by guidelines on uncertainty calculation in such complex situations, and could be more widely used by the community.

The article is generally well written and organised. The main criticism would be a lack of clarity in the assumptions made by the authors when using datasets published by other teams, and in applying the Monte-Carlo simulations, as detailed in more specific comments below:

Specific comments:

1. The introduction could give more details on the needs. It is explained that remote sensing applications use the reviewed datasets. However this paper is about their uncertainty, and it is not said how this uncertainty is or would be used. It is important because the authors present a good piece of work with the Monte-Carlo simulations, which would not be valuable if users could be satisfied for example with just a conservative relative uncertainty to be applied to all values. This may be straightforward to the authors, but should be developed in the paper.

2. The choice of Monte-Carlo simulation could also be developed already in the introduction. Why this methodology? What are the assumptions? What does it bring? Could it be used for other cases?

3. Section 2 – measurement technique could be better organised. It first introduces the common aspect of all measurements of the cross-section, and then mentions some particularities. This could be introduced, explaining the sources of uncertainties that are expected to be always present due to the Beert-Lamber law, and then additional sources due to experimental choices.

4. Section 3 - review of reported uncertainty is interesting but the reader misses a clear goal. It would be easier to read having in mind the purpose of that review. Is this to provide the inputs for the Monte-Carlo simulation? In that case, why separating type A and type B uncertainties? Monte-Carlo calculation of uncertainties does not make such a distinction. It just needs the PDF associated with all uncertainty sources. One could also think that the users of data sets would need to know which uncertainties are correlated. This would be valuable but this point is never clearly mentioned in that section.

5. Section 3.1 uncertainty budget of Hearn: the goal of that section is unclear, and this subject was already treated in a publication by Viallon *et al.* in 2006 (DOI: 10.1088/0026-1394/43/5/016). It is certainly relevant to note that the BP data set is scaled to the Hearn value, and that BP uncertainties should be at least that of Hearn. However this could be stated shortly within section 3.2. In addition the review of Hearn uncertainty budget contains some inconsistencies and its conclusion is unclear (see line by line comments). It is suggested to either reconsider the analysis or remove this section.

6. Section 4 also needs to clarify the uncertainty treatment associated with the polynomial fit. How are the uncertainties of the polynomial coefficients calculated? Are they outputs of the regressions? If that is the case, what are the uncertainties taken into account in the regression? None, which would justify the move towards Monte-Carlo? But then, what about trying to just use the reported uncertainties associated with each point and perform the regression with those? What would be the difference with Monte-Carlo simulations?

7. Section 5 leaves unanswered questions regarding the Monte-Carlo calculations. Based on the guide cited by the authors (JCGM 101:2008), one would expect some considerations on the measurement equation or process, identifying the input quantities, and explaining the choice of the PDFs associated with each of them. In this paper it is difficult to link uncertainties accounted for in the Monte-Carlo simulation, listed in Table 8, and the input quantities. It is not clear if authors have considered just equation 4 (polynomial) as their model, or a more complex process involving several steps of fitting. In addition, Monte-Carlo simulations to calculate uncertainties implement different version of an algorithm, with different assumptions. Reference to the programme (if external) or the code (if authors did the programme) is missing here. Reference to Wu 1986 found in section 5 seems to be on least-square analysis rather than Monte-Carlo calculations. This is confusing, in particular when the problem seems to be solvable by a least-square code.

8. Validation of the Monte-Carlo method: authors do not mention how they have validated their choice of a simulation package. Have they looked at a few measured cross-sections to compare the output of MCM with the measurement results uncertainty?

9. Section 6 nicely summarises the work done. However conclusions drawn by the authors could go beyond this work. Some comments on the current usage of those datasets, highlighting how uncertainties are commonly neglected, would bring added value to the current study. If the goal is to encourage more careful consideration of the uncertainty associated with those datasets, this should be better emphasized.

10. The authors are kindly asked to review how to express units and quantities. They can refer to the SI brochure for example (mostly available on-line) or the ISO guide 80000-1:2009. Unit names for instance are normally written in roman (upright) type, and they are treated like ordinary nouns. Symbols for quantities are generally single letters set in an italic font.

Line-by-line comments:

Line 28: "…the satellite community uses any of the three data sets or other data". Which other data?

Line 32: "Neither of the publications uses the guidelines…". This sounds a bit unfair. Most of the work was published before the JCGM guidelines. The concept of uncertainty was already there, but the vocabulary was not harmonised, and indeed some uncertainty sources were underestimated.

Line 70: the standard deviation of the mean can indeed be used as standard uncertainty in case of white noise, which is often the case in spectroscopic instruments. However what is the implication of this statement? In published papers uncertainty of the mean was often an experimental standard deviation. Do the authors suggest this should be replaced with the standard deviation of the mean?

Line 101-104: what is the rationale behind the choice of a rectangular distribution for the uncertainty published by Hearn? This does not seem consistent with table 1.

Line 104: the term precision brings more confusion than clarity. Is that an uncertainty? What is finally the uncertainty on the Hearn value considered by the authors?

Line 119: "it seems to be an underestimation". Agreed, but what is the decision taken by the authors?

Line 119: it is here stated that the Hearn value is reported with a 1.4% relative standard uncertainty. This was not clear from section 3.1, and is arguable as already noticed. As Hearn provided an uncertainty budget, it seems more reasonable to revise it than using his value with a rectangular distribution of the uncertainty.

Line 153: please explain the "spectral registration". For non-expert, this appears to be a sort of shift applied to the wavelength scale. Is that more complex? How does this process create a statistical uncertainty?

Line 157 "standard deviation (variances)". The variance being the square of the deviation, what is the meaning of the bracket? To clarify that the calculation of uncertainty is performed with the variance, or to state that the standard uncertainty associated with the intensity is the experimental standard deviation? Both? Please clarify.

Line 192 equation (4): the choice of the symbol $t$ for the temperature in °C is rather unusual, as this is normally the time. More important, it may be more robust to always use the temperature in K. it is not clear why this change is needed.

Line 193 " multivariate linear regression". Why *linear* when the cross-section dependency versus the temperature is a second order polynomial?

Line 197: use GUM convention in equation (5): $u^2(\sigma_p) = u^2(a_0) + t^2 u^2(a_1) + t^4 u^2(a_2)$. This equation also assumes that the uncertainty in the temperature is negligible. This should be stated and motivated.

Line 207-212: it seems rather "unfair" for other groups to choose a temperature of 193 K to compare results, as only SG performed actual measurements at that temperature. It may not change the conclusion, but it is suggested to consider another temperature.

Line 213-215: please clarify the purpose of this statement. It is currently difficult to know what to do with it.

Line 222: "..only the uncertainty from the temperature parametrisation using a polynomial". Does this mean that no experimental uncertainty was included? This should be stated more clearly, and some justification provided for not doing it (in section 4).

Line 253-260: analysis of results: could be put in perspective with uncertainties reported in the literature and choices made for this paper. For instance, the 2% uncertainty on BP certainly reflects the uncertainty from using Hearn value as reference.

Table 1: in Hearn paper, the column "Total SD (RMS)" is simply the combined relative standard uncertainty, calculated from sources listed in other columns. This is consistent on all wavelengths measured by Hearn. This is badly reflected in Table 1.

Table 5: this table tends to associate statistical uncertainty with Type A, and systematic with type B. As pointed earlier in the paper, this is a confusion that should be avoided. Type B uncertainties in particular may not be systematic (or biases), but all uncertainties for which the information was not provided by repeated observations (calibration certificate for instance). It may be the case in the reported experiment that all Type B uncertainties are also systematic. It is then suggested to modify the titles to reflect the type (A or B). However again, one wonders why the distinction is made, when it is no use in Monte-Carlo simulations.

Figures 2 to 4: the information displayed in those figures is too raw, and does not help the reader in analysing the results. It is impossible to compare the 3 plots, difficult to see the level of uncertainties. The choice of those figures should be questioned. Figure 5 only seems to be sufficient in that section.

Figure 4.b) This graph is too busy. What is the goal? If this is to show that a second order polynomial was appropriate, then statistical tools can be used, such as residuals standard deviations. If the authors want to provide the ranges of residuals for each wavelength in the graph, then a table might be more appropriate.

Editorial/technical corrections:

Line 92: O3 and O2 to be written $O_3$ and $O_2$.

Line 136: avoid the symbol @ to mean "at".

Table 1: the molecular absorption cross-section value misses a multiplication sign and a power notation: $2.4 \times 10^{-19}$.

Table 2: the uncertainty in the transmittance determination misses a power notation: "2 in $10^5$" and not "2 in 105".

Table 3: avoid the symbol @ in place of "at"

Figures 2 to 4: axis titles are incomplete: $x$ axis should be "$\lambda$ / nm" with the explanation in the legend that $\lambda$ is the wavelength.

---

## Referee Comment (RC2) · Anonymous Referee #2 · 13 Jun 2016

This paper presents a review of the uncertainty budgets of three important ozone absorption cross-section datasets. This is a valuable exercise with a very practical application, since these uncertainties should be included in uncertainty estimates of ozone measurements made in the field by ground-based UV absorption monitors and by total ozone spectrophotometers, both ground and satellite-based. However, it will be much more valuable if it is widely used, and in that respect the paper would benefit greatly from some clear explanation, with examples, of the application of these results (which we are told are available on the authors' web site) to standard ozone monitoring and reporting. Some more detail in the description of what has been done in this paper would also help (perhaps in a supplement). The authors are reminded of the principle that a paper should describe the analysis in sufficient detail that the reader could reproduce it, if s/he so chose.

Detailed comments:

Section 2: The first paragraph, introducing the Beer-Lambert law, would seem to fit better in the Introduction.

Lines 55-59 and 60-64: Although the second of these paragraphs (line 60) begins with "Alternatively...", I do not understand how these two situations differ. It appears that both scale their measurements to other published ozone absorption cross-sections?

Lines 79-80: Why does concatenating spectra lead to additional uncertainties? Are those uncertainties not already accounted for in the bias uncertainties of the individual "slices"? (See also lines 156-158, below).

Lines 101-102: Why do you assume a rectangular distribution? In the absence of other information, would it not be more conservative to treat the reported value as a standard error, and assume a Gaussian distribution?

Line 128: "...extend the wavelength coverage (into the visible)..."? The wavelength ranges quoted differ by only 2 nm at the longer end.

Table 3: I have difficulty seeing how the individual uncertainties quoted add up to the totals at the bottom.

Lines 151-158: Some more detail here, please! From the lower panel of Figure 1, I gather that the relative uncertainties increase generally with an exp(lambda^2) dependence on either side of a central wavelength in each "slice", but some other term is dominating at the longer wavelength end, and it is not at all obvious why the errors should take this form. The sentence "The latter were calculated according to law of propagation of uncertainty using standard deviation (variances) and mean values of I and I0 spectra which determine the optical density OD (see Eq. 3)." is no help in this regard.

Figures 2-4: These need better labels, and captions. Specifically, how are the uncertainties shown? There appears to be a shaded range in Figure 3, but I can't see

any in Figure 2, although Figure 5 suggests that the BP temperature uncertainties are much larger. I also suggest removing the sigma(T) equation from the figure, to avoid ambiguity with a0.

Lines 207-212: The uncertainties for the BP are an order of magnitude larger in Figure 5 than for the other cross-section datasets. This contrasts remarkably with Figure 2, but the authors do not remark on this apparent contradiction, nor explain it. Probably what is needed here is more description of the calculations, and possibly a few supplementary figures.

Figures 6 and 7: It would be nice to see a fourth panel superimposing the three others, in these two figures. It looks like the three cross-section datasets agree pretty well above 220K.

Summary: There is a lot more that could be said here. What is the practical effect of the community's use of these different ozone absorption cross-section datasets? The authors are presumably well-equipped to recalculate a few selected ozone measurements under different atmospheric conditions, and show how they differ both in mean and uncertainty, using the three cross-section datasets. This would make the paper more immediately interesting by demonstrating the practical importance of this work.

---

## Author Comment (AC1) · 20 Jul 2016

**Reply to reviewer comments on AMTD paper "Uncertainty budgets of major ozone absorption cross-sections used in UV remote sensing applications" by Mark Weber, Victor Gorshelev, and Anna Serdyuchenko**

**Note:** Reviewer statements are in italic and our responses are in blue below the comments

**Reviewer 1:**

*General comments: The submitted article presents a critical review of three published datasets of ozone absorption cross-sections in the Huggins band, including data previously published by the same authors, with a focus on their uncertainty. In that regard it addresses an issue which has often been underestimated. It should bring valuable outputs for the community of scientists involved in ozone monitoring with remote sensing instruments, in providing a sound base to select the most appropriate dataset to finally improve the confidence in measurements. To calculate uncertainties associated with cross-sections at any temperature, the authors have chosen to use Monte-Carlo simulations. This tool is recommended by guidelines on uncertainty calculation in such complex situations, and could be more widely used by the community.*

*The article is generally well written and organised. The main criticism would be a lack of clarity in the assumptions made by the authors when using datasets published by other teams, and in applying the Monte-Carlo simulations, as detailed in more specific comments below:*

Thanks for the kind judgement. We addressed the specific comments in our reply below.

 *Specific comments:*

1. *The introduction could give more details on the needs. It is explained that remote sensing applications use the reviewed datasets. However this paper is about their uncertainty, and it is not said how this uncertainty is or would be used. It is important because the authors present a good piece of work with the Monte-Carlo simulations, which would not be valuable if users could be satisfied for example with just a conservative relative uncertainty to be applied to all values. This may be straightforward to the authors, but should be developed in the paper.*
We have extended the introduction (see our reply to the next point). This work is part of a project on traceability of total ozone measurements (ATMOZ), where we plan several applications of our results in simulated retrievals as well as the general impact on satellite and ground retrievals of total ozone. This will be subject of other publications.

2. *The choice of Monte-Carlo simulation could also be developed already in the introduction. Why this methodology? What are the assumptions? What does it bring? Could it be used for other cases?*
The introduction has been expanded (second to last paragraph) by adding the following: "In this paper we use a Monte Carlo simulation in order to get a better estimate of uncertainties

in the major ozone absorption cross-section data. This method will allow us to combine the uncertainties from the laboratory measurements with those from the temperature parameterisation that is widely used to interpolate between the laboratory temperatures. The temperature dependence in the ozone absorption cross section is usually expressed as a quadratic polynomial (Bass-Paur parameterisation). Having only four to five temperatures available as in the case of BDM and BP a quadratic polynomial will at many wavelengths overfit the data (perfect matches). However, in MC simulation that includes uncertainties from the laboratory measurements more realistic uncertainties can be obtained also for the temperature parameterisation."

3.  *Section 2 – measurement technique could be better organised. It first introduces the common aspect of all measurements of the cross-section, and then mentions some particularities. This could be introduced, explaining the sources of uncertainties that are expected to be always present due to the Beer-Lambert law, and then additional sources due to experimental choices.*
    Agreed, changes have been made to better introduce the issue. After Eq. 3 we added the following: "The Beer-Lambert law, used to describe the absorption, is applicable when the density of the absorbing medium is low enough to avoid non-linear effects, medium itself does not scatter light, and exposure to light does not change the properties of the medium. The first two points are valid for gaseous media like the ozone sample collected in the experimental cell. Exposure to UV radiation leads to photolysis of ozone and causes an observable change in ozone concentration if measurement time is long enough… Uncertainties of cross-sections derived from Eq. 3 are influenced by measurement uncertainties of these contributing parameters. "

4.  *Section 3 - review of reported uncertainty is interesting but the reader misses a clear goal. It would be easier to read having in mind the purpose of that review. Is this to provide the inputs for the Monte-Carlo simulation? In that case, why separating type A and type B uncertainties? Monte-Carlo calculation of uncertainties does not make such a distinction. It just needs the PDF associated with all uncertainty sources. One could also think that the users of data sets would need to know which uncertainties are correlated. This would be valuable but this point is never clearly mentioned in that section.*
    This is correct that for the MC simulation only systematic and random components are relevant (not type A/B uncertainties). We have corrected this in this section. We assume here that the uncertainties between adjacent wavelengths are uncorrelated. We added a statement in the new extended description of the MC simulation in  Section 5. The modified text reads now: " Uncertainties for all parameters in Table 8 are drawn from a Gaussian random generator to perturb the cross section data at each available temperature. Random uncertainties means that for each temperature available a new set of random perturbations were calculated, while systematic uncertainties means uncertainties drawn from the random generator were applied to all temperature data simultaneously. A total of 10,000 perturbed datasets of cross section data were then generated and each fitted by a quadratic polynomial in temperature. The $1\sigma$ distributions from the sample polynomials provided then the overall $1\sigma$ uncertainty as function of temperature. MC simulations were repeated for all

5. *Section 3.1 uncertainty budget of Hearn: the goal of that section is unclear, and this subject
was already treated in a publication by Viallon et al. in 2006 (DOI: 10.1088/0026-
1394/43/5/016). It is certainly relevant to note that the BP data set is scaled to the Hearn
value, and that BP uncertainties should be at least that of Hearn. However this could be
stated shortly within section 3.2. In addition the review of Hearn uncertainty budget contains
some inconsistencies and its conclusion is unclear (see line by line comments). It is suggested
to either reconsider the analysis or remove this section.*
A brief summary of the uncertainties reported by (Hearn, 1961) seems to be relevant since
later BP data is scaled to Hearn data.

6. *Section 4 also needs to clarify the uncertainty treatment associated with the polynomial fit.
How are the uncertainties of the polynomial coefficients calculated? Are they outputs of the
regressions? If that is the case, what are the uncertainties taken into account in the
regression? None, which would justify the move towards Monte-Carlo? But then, what about
trying to just use the reported uncertainties associated with each point and perform the
regression with those? What would be the difference with Monte-Carlo simulations?*
In Section 4 only uncertainties from the polynomial fits (ignoring the measurement
uncertainties) are reported. Since for some data only a limited number of temperatures are
available (four to five) the polynomial fits are sometimes perfect (overfitting, zero
uncertainty). The use of the MC simulation combining measurement uncertainties and
polynomial uncertainties allows a better assessment of the overall uncertainties. This is
made now clearer in the introduction and also Section 4.

7. *Section 5 leaves unanswered questions regarding the Monte-Carlo calculations. Based on the
guide cited by the authors (JCGM 101:2008), one would expect some considerations on the
measurement equation or process, identifying the input quantities, and explaining the choice
of the PDFs associated with each of them. In this paper it is difficult to link uncertainties
accounted for in the Monte-Carlo simulation, listed in Table 8, and the input quantities. It is
not clear if authors have considered just equation 4 (polynomial) as their model, or a more
complex process involving several steps of fitting. In addition, Monte-Carlo simulations to
calculate uncertainties implement different version of an algorithm, with different
assumptions. Reference to the programme (if external) or the code (if authors did the
programme) is missing here. Reference to Wu 1986 found in section 5 seems to be on least-
square analysis rather than Monte-Carlo calculations. This is confusing, in particular when
the problem seems to be solvable by a least-square code.*
In the MC simulation the cross-section data at the available temperatures are perturbed
using Gaussian random distributed uncertainties from Table 8. A total of 10000 simulated
cross-section data were each fitted by a quadratic polynomial and the 1sigma distribution
of the polynomials then provides the overall uncertainty. Only quadratic fits were used. As
expected the mean value of all polynomials in the MC simulation agrees with the polynomial
fitted to the unperturbed cross-section data. So the MC simulation also includes least

squares approaches. The explanation of the MC simulation has been extended to make this more clear (see answer to point 4 above).

8. *Validation of the Monte-Carlo method: authors do not mention how they have validated their choice of a simulation package. Have they looked at a few measured cross-sections to compare the output of MCM with the measurement results uncertainty?*
We have not directly validate the MC simulation, but from Figures 6 and 7, the uncertainties from the laboratory measurements are indicated by the red crosses which show the overall uncertainty from the laboratory measurements (systematic and random, vertical bar) as well as the temperature uncertainty (horizontal bar). The overall uncertainties are generally in good agreement with the MC total uncertainty estimate, however, Figure 6a (BP at 319.4 nm) clearly indicates that the overall uncertainty is here larger than the uncertainties from the measurements and this is due to the outlier at 203 K which increases largely the contribution from the uncertainty of the polynomial. In the discussion on Figs. 6 and 7 this is now mentioned.

9. *Section 6 nicely summarises the work done. However conclusions drawn by the authors could go beyond this work. Some comments on the current usage of those datasets, highlighting how uncertainties are commonly neglected, would bring added value to the current study. If the goal is to encourage more careful consideration of the uncertainty associated with those datasets, this should be better emphasized.*
We added the following to emphasisze this. "In this paper we attempted to provide a more realistic uncertainty budget that may be useful when trying to establish the contribution from ozone absorption cross-sections to the overall uncertainty of retrieved ozone. This work is part of a project on traceability of total ozone measurements (ATMOZ), where we plan several applications of our results in simulated retrievals as well as the general impact on satellite and ground retrievals of total ozone. This will be subject of other publications*.

10. *The authors are kindly asked to review how to express units and quantities. They can refer to the SI brochure for example (mostly available on-line) or the ISO guide 80000-1:2009. Unit names for instance are normally written in roman (upright) type, and they are treated like ordinary nouns. Symbols for quantities are generally single letters set in an italic font.*
This will be taken care by the typesetting of the final manuscript. All units are now written upright.

**Line-by-line comments:**

*Line 28: "…the satellite community uses any of the three data sets or other data". Which other data?*

We changed to "while the satellite community uses any of the three data sets when retrieval is limited to the UV spectral region." Satellites that also employ visible and near IR wavelengths use other cross-section data as BDM only covers parts of the visible and BP is only available in the UV below 340 nm.

*Line 32: "Neither of the publications uses the guidelines…". This sounds a bit unfair. Most of the work was published before the JCGM guidelines. The concept of uncertainty was already there, but the vocabulary was not harmonised, and indeed some uncertainty sources were underestimated.*

Agreed. Omit the sentence.

*Line 70: the standard deviation of the mean can indeed be used as standard uncertainty in case of white noise, which is often the case in spectroscopic instruments. However what is the implication of this statement? In published papers uncertainty of the mean was often an experimental standard deviation. Do the authors suggest this should be replaced with the standard deviation of the mean?*

It simply means here that repeated measurements not only reduce SNR but also includes uncertainty from potential lamp drifts. We added "then" to this sentence, so that this applies to repeated measurements.

*Line 101-104: what is the rationale behind the choice of a rectangular distribution for the uncertainty published by Hearn? This does not seem consistent with table 1.*

Agreed, assumption of the normal distribution is more appropriate. Changes to the manuscript will be added.

*Line 104: the term precision brings more confusion than clarity. Is that an uncertainty? What is finally the uncertainty on the Hearn value considered by the authors?*

Agreed. We changed the sentence to "Adding up all uncertainties the Hearn value has an overall uncertainty of2%"

*Line 119: "it seems to be an underestimation". Agreed, but what is the decision taken by the authors?*

We added the following here: "We later assume an systematic uncertainty of about 2% (systematic) which is somewhat higher than the 1.3% estimated for DBM and SG (see next two subsections)."

*Line 119: it is here stated that the Hearn value is reported with a 1.4% relative standard uncertainty. This was not clear from section 3.1, and is arguable as already noticed. As Hearn provided an uncertainty budget, it seems more reasonable to revise it than using his value with a rectangular distribution of the uncertainty.*

see reply to previous item

*Line 153: please explain the "spectral registration". For non-expert, this appears to be a sort of shift applied to the wavelength scale. Is that more complex? How does this process create a statistical uncertainty?*

registration is here the wrong word, we meant "spectral measurements"

*Line 157 "standard deviation (variances)". The variance being the square of the deviation, what is the meaning of the bracket? To clarify that the calculation of uncertainty is performed with the variance, or to state that the standard uncertainty associated with the intensity is the experimental standard deviation? Both? Please clarify.*

We omit "(variances)"

*Line 192 equation (4): the choice of the symbol t for the temperature in °C is rather unusual, as this is normally the time. More important, it may be more robust to always use the temperature in K. it is not clear why this change is needed.*

The temperature parameterisation is commonly expressed in degree C, where we use the small letter t (Capital T for temperatures in K).

*Line 193 " multivariate linear regression". Why linear when the cross-section dependency versus the temperature is a second order polynomial?*

Polynomial fits are special cases of multivariate linear regressions. They are called linear as the equation is linear with respect to the coefficients (to be fitted).

*Line 197: use GUM convention in equation (5): This equation also assumes that the uncertainty in the temperature is negligible. This should be stated and motivated.*

AMT is not following GUM conventions and usually writes the equation as is and there is no ambiguity here. Leave as is.

*Line 207-212: it seems rather "unfair" for other groups to choose a temperature of 193 K to compare results, as only SG performed actual measurements at that temperature. It may not change the conclusion, but it is suggested to consider another temperature.*

Here we make the point that extrapolation of cross-section data beyond the temperature range of the laboratory measurements leads to quite large uncertainties. The advantage of the MC simulation is that uncertainties for all temperatures of interest are available.

*Line 213-215: please clarify the purpose of this statement. It is currently difficult to know what to do with it.*

We removed the sentence "The instrumental slit function can be easily applied ..." with "In order to determine the ozone cross-section at a specific instrument resolution one can either convolve all the various temperature data with the instrument function and then apply the polynomial fit or the coefficient spectra (as shown in Figs. 2-4) are convolved and the polynomial coefficients from the original data are used." This makes it clearer.

*Line 222: "..only the uncertainty from the temperature parametrisation using a polynomial". Does this mean that no experimental uncertainty was included? This should be stated more clearly, and some justification provided for not doing it (in section 4).*

The uncertainties given in Section 4 (Eq. 5) only account for uncertainties in the polynomial fit, excluding all other experimental uncertainties. We extended the first sentences in this paragraph as follows: "The uncertainty given in Eq. 5 reflects only the uncertainty from the temperature parameterisation using a polynomial (if we assume that a quadratic dependence in temperature is true), thus excluding the experimental uncertainties as discussed in Section 3. One main motivation to only show the uncertainties arising from the polynomial fit is to demonstrate that with only few

temperatures available for some of the datasets (BP, DBM) the uncertainty in the temperature dependence is strongly underestimated due to overfitting."

*Line 253-260: analysis of results: could be put in perspective with uncertainties reported in the literature and choices made for this paper. For instance, the 2% uncertainty on BP certainly reflects the uncertainty from using Hearn value as reference.*

We modified the following sentence by adding the part marked with <>: "The larger systematic measurement uncertainty of the BP data (2%)<, due to the uncertainties related to the Hearn value at the mercury line used for scaling the BP data (see Section 3.1),> leads to larger overall uncertainties in the BP data.

*Table 1: in Hearn paper, the column "Total SD (RMS)" is simply the combined relative standard uncertainty, calculated from sources listed in other columns. This is consistent on all wavelengths measured by Hearn. This is badly reflected in Table 1.*

Agreed, the table will be corrected to properly represent the data.

*Table 5: this table tends to associate statistical uncertainty with Type A, and systematic with type B. As pointed earlier in the paper, this is a confusion that should be avoided. Type B uncertainties in particular may not be systematic (or biases), but all uncertainties for which the information was not provided by repeated observations (calibration certificate for instance). It may be the case in the reported experiment that all Type B uncertainties are also systematic. It is then suggested to modify the titles to reflect the type (A or B). However again, one wonders why the distinction is made, when it is no use in Monte-Carlo simulations.*

Here we report on systematic and random uncertainties not type A and B. We changed the table accordingly.

*Figures 2 to 4: the information displayed in those figures is too raw, and does not help the reader in analysing the results. It is impossible to compare the 3 plots, difficult to see the level of uncertainties. The choice of those figures should be questioned. Figure 5 only seems to be sufficient in that section.*

We tried different plots (e.g. comparing directly the coefficients) but figures get to crowdy. Suggest to leave as is as they show some pecularities of the various datasets. We also added the following description to Figs. 2-4 in the main text: "These figures also show that the BP data appear somewhat noisier than the others and one striking difference between SG and BDM is the apparent bump in the third coefficient (blue line) near 305 nm only evident in BDM."

*Figure 4.b) This graph is too busy. What is the goal? If this is to show that a second order polynomial was appropriate, then statistical tools can be used, such as residuals standard deviations. If the authors want to provide the ranges of residuals for each wavelength in the graph, then a table might be more appropriate.*

We disagree here. A table cannot show the spectral behavior as at some wavelengths the uncertainties strongly varies in a narrows spectral range.

*Editorial/technical corrections:*

*Line 92: O₃ and O₂ to be written O$_3$ and O$_2$.*

done

*Line 136: avoid the symbol @ to mean "at".*

done

*Table 1: the molecular absorption cross-section value misses a multiplication sign and a power notation: $2.4 \times 10^{-19}$.*

done

*Table 2: the uncertainty in the transmittance determination misses a power notation: "2 in 10$_5$" and not "2 in 105".*

done

*Table 3: avoid the symbol @ in place of "at"*

done

*Figures 2 to 4: axis titles are incomplete: x axis should be "λ / nm" with the explanation in the legend that λ is the wavelength.*

done

**Reviewer 2:**

*This paper presents a review of the uncertainty budgets of three important ozone absorption cross-section datasets. This is a valuable exercise with a very practical application, since these uncertainties should be included in uncertainty estimates of ozone measurements made in the field by ground-based UV absorption monitors and by total ozone spectrophotometers, both ground and satellite-based. However, it will be much more valuable if it is widely used, and in that respect the paper would benefit greatly from some clear explanation, with examples, of the application of these results (which we are told are available on the authors' web site) to standard ozone monitoring and reporting. Some more detail in the description of what has been done in this paper would also help (perhaps in a supplement). The authors are reminded of the principle that a paper should describe the analysis in sufficient detail that the reader could reproduce it, if s/he so chose.*

See also our reply  to specific comment #1 from R#1. We have extended the introduction to motivate this work better. We added in the summary the following: "This work is part of a project on traceability of total ozone measurements (ATMOZ), where we plan several applications of our results in simulated retrievals as well as the general impact on satellite and ground retrievals of total ozone. This will be subject of other publications."

*Detailed comments:*

*Section 2: The first paragraph, introducing the Beer-Lambert law, would seem to fit better in the Introduction.*

We think it better fits in the beginning of Section 2

*Lines 55-59 and 60-64: Although the second of these paragraphs (line 60) begins with "Alternatively...", I do not understand how these two situations differ. It appears that both scale their measurements to other published ozone absorption cross-sections?*

Agreed, we start the sentence ("Alternatively, measurements ...") now as follows: "Absolute measurements performed at selected wavelengths ...are often used ..."

*Lines 79-80: Why does concatenating spectra lead to additional uncertainties? Are those uncertainties not already accounted for in the bias uncertainties of the individual "slices"? (See also lines 156-158, below).*

If all the experimental data necessary for the absolute scaling of individual OD spectra slices was available (temperature, absorption path length and ozone number density derived from pressure measurement), then no additional uncertainty is introduced.

If, however, measurement conditions during spectra acquisition did not allow for reliable measurements of the necessary parameter (typically it is the absolute ozone pressure), then the concatenation of the two adjacent spectra slices is performed by fitting one slice to another within certain small (typically, several nanometers) wavelength range where both spectra have reliable individual measurement uncertainty. The fitting procedure is based on a RMS linear fit, which has a certain intrinsic uncertainty.

*Lines 101-102: Why do you assume a rectangular distribution? In the absence of other information, would it not be more conservative to treat the reported value as a standard error, and assume a Gaussian distribution?*

Agreed, assumption of the normal distribution is more appropriate.

*Line 128: "...extend the wavelength coverage (into the visible)..."? The wavelength ranges quoted differ by only 2 nm at the longer end.*

DBM data cover up to 520 nm or more dependent on the temperature. This has been corrected in the main text and Table 7

*Table 3: I have difficulty seeing how the individual uncertainties quoted add up to the totals at the bottom.*

In its current form, Table 3 is the combination of data from the two tables titled "Table II. Analysis of errors" in (Daumont et al., 1992) and (Malicet et al., 1995). Below is the side-by-side view of the data from the original tables:

| | Daumont et al., (1992) | Malicet et al., (1995) |
|---|---|---|
| Quantity | Uncertainty | Uncertainty |

| | | |
|---|---|---|
| Absorbance (for λ < 335 nm) | 1 % | 1 % |
| Optical path | 0.05 % | 0.05 % |
| Ozone pressure | 0.1 % | 0.1 % |
| Impurities | < 0.1 % | < 0.1 % |
| Temperature | 0.02 % | from 0.02 % at 295 K up to 0.15 % at 218 K |
| Wavelength | < 0.05 % (Hartley band) < 0.8 % (Huggins band) | 0.005-0.015 nm |
| Total (systematic) error | 1.3 % (Hartley band) 1.3 – 2.5 % (Huggins band) | 1.3 – 1.5 % (Hartley band) 1.3 – 3.5 % (Huggins band) |
| Random error (RMS) (for λ< 340 nm) | 0.9 – 2.2 % | 0.3 – 2.0 % |

Both publications are a bit vague on the subject of detailed uncertainty classification and budget and seem to use a slightly differing approach for calculation of the "Total (systematic) error" and "Random errors (RMS)". We decided to replace Table 3 with this one.

*Lines 151-158: Some more detail here, please! From the lower panel of Figure 1, I gather that the relative uncertainties increase generally with an exp(lambdaˆ2) dependence on either side of a central wavelength in each "slice", but some other term is dominating at the longer wavelength end, and it is not at all obvious why the errors should take this form. The sentence "The latter were calculated according to law of propagation of uncertainty using standard deviation (variances) and mean values of I and I0 spectra which determine the optical density OD (see Eq. 3)." is no help in this regard.*

As a matter of fact, the relative uncertainty of OD (or X-sections) within the slice should be a function inversely proportional to the OD (or X-section).

To understand why the relative uncertainty demonstrates such a behaviour, one needs to consider the mean spectra for Io and I. The stability of the light source causes more or less flat variance distribution with wavelength for both Io and I.

The absorption cross-section decreases with wavelength between 260-380 nm, meaning that the intensity I registered by the spectrometer on the shorter wavelength end of the given range, where absorption is strong, is either very small or close to baseline ("dark current") signal of the instrument. At the same time, both Io and I spectra have a very similar intensity at the longer wavelength end of the slice.

So for calculation of OD from Io and I within every slice two issues are present:

1) Io and I are very similar spectra with equal influence of the spectrometer detector noise (longer wavelength end of the slice)

2) Io and I are dramatically different spectra, one being close to the baseline signal of the instrument (shorter wavelength end of the slice)

It is the influence of the spectrometer detector noise and resulting non-linearities in intensity registration that produces the artefacts on the shorter wavelength end of the slice. This is one of the reasons why ODs from only certain range (approx. 0.1-1) are considered for further processing.

*Figures 2-4: These need better labels, and captions. Specifically, how are the uncertainties shown? There appears to be a shaded range in Figure 3, but I can't see any in Figure 2, although Figure 5 suggests that the BP temperature uncertainties are much larger. I also suggest removing the sigma(T) equation from the figure, to avoid ambiguity with a0.*

Figures have been changed (see comments by reviewer 1), however, for some wavelengths and coefficients the uncertainty is so small that they are not visible. We added in the figure caption: "For some coefficients and/or selected wavelengths the uncertainties are not visible."

*Lines 207-212: The uncertainties for the BP are an order of magnitude larger in Figure 5 than for the other cross-section datasets. This contrasts remarkably with Figure 2, but the authors do not remark on this apparent contradiction, nor explain it. Probably what is needed here is more description of the calculations, and possibly a few supplementary figures.*

We do not see a contradiction as Figs. 2-5 shows only the uncertainties of the individual coefficients, but when calculating the total uncertainties of the cross-sections from the polynomial then the values are multiplied with t or t^2, so the figures cannot be directly compared.

*Figures 6 and 7: It would be nice to see a fourth panel superimposing the three others,  in these two figures. It looks like the three cross-section datasets agree pretty well above 220K.*

*We tried to superimpose the three datasets, but the figure gets to crowded and hard to read.*

**Summary:** *There is a lot more that could be said here. What is the practical effect of the community's use of these different ozone absorption cross-section datasets? The authors are presumably well-equipped to recalculate a few selected ozone measurements under different atmospheric conditions, and show how they differ both in mean and uncertainty, using the three cross-section datasets. This would make the paper more immediately interesting by demonstrating the practical importance of this work.*

See our reply to specific comment 9 of Reviewer 1. We are preparing a detailed study on its impact using simulated retrievals and retrieval applications to satellite and ground data as part of the EMRP ATMOZ project. This will be subject of other papers and, thus, is beyond the scope of this paper.

**Attachment:**

Revised manuscript with track changes

**Uncertainty budgets of major ozone absorption cross-sections used in UV remote sensing applications**

Mark Weber[1], Victor Gorshelev[1], and Anna Serdyuchenko[1*]

[1]Institut für Umweltphysik, Universität Bremen FB1, PO Box 330 440, D-28334 Bremen, Germany

*now at: OHB System AG, Manfred-Fuchs-Straße 1, D-82234 Weßling, Germany

*Correspondence to*: Mark Weber (weber@uni-bremen.de)

**Abstract.** Detailed uncertainty budgets of three major UV ozone absorption cross-section datasets that are used in remote sensing application are provided and discussed. The datasets are Bass-Paur (BP), Brion-Daumont-Malicet (BDM), and the more recent Serdyuchenko-Gorshelev (SG). For most remote sensing application the temperature dependence of the Huggins ozone band is described by a quadratic polynomial in temperature (Bass-Paur parameterisation) by applying a regression to the cross-section data measured at selected atmospherically relevant temperatures. For traceability of atmospheric ozone measurements uncertainties from the laboratory measurements as well as from the temperature parameterisation of the ozone cross-section data are needed as an input to detailed uncertainty calculation of atmospheric ozone measurements. In this paper the uncertainty budgets of the three major ozone cross-section datasets are summarised from the original literature. The quadratic temperature dependence of the cross-section datasets is investigated. Combined uncertainty budgets is provided for all data sets based upon Monte Carlo simulation that includes uncertainties from the laboratory measurements as well as uncertainties from the temperature parameterisation. Between 300 and 330 nm both BDM and SG have an overall uncertainty of 1.5%, while BP has a somewhat larger uncertainty of 2.1%. At temperatures below about 215 K, uncertainties in the BDM data increase more strongly than the others due to the lack of very low temperature laboratory measurements (lowest temperature of BDM available is 218 K).

**1 Introduction**

The three ozone absorption cross-sections in common use for many remote sensing applications are the Bass Paur (BP) data (Bass and Paur 1985; Paur and Bass, 1985), the Daumont-Brion-Malicet (BDM) data (Daumont et al., 1992; Brion et al., 1993; Malicet et al., 1995) and the very recent Serdyuchenko-Gorshelev (SG) data (Gorshelev et al., 2014; Serdyuchenko et al., 2011; 2014). While the data from BDM and SG are absolute cross-section measurements, the BP data were scaled to the so-called Hearn value at the mercury line wavelength (253.65 nm). The standard retrievals applied to the ground Brewer and Dobson spectrophotometer data use the BP data (e.g. Redondas et al., 2014), while the satellite community uses any of the three data sets when retrieval is limited to the UV spectral region  (WMO-GAW, 2015; ACSO, 2010; Orphal et al., Absorption cross-sections of ozone - Status report 2015, manuscript in preparation).

30       For the review of uncertainties, original publications reporting on results of the experimental work were considered first. Since BP data was absolutely scaled using Hearn data (Hearn, 1961), the latter is also included in this review. There is a lack of consistency in the presentation of measurement uncertainty budgets across different papers.  An attempt to analyze and harmonize the reported uncertainties is made in the following sections. In some cases not all measured quantities were reported and in

35  most cases detailed description of the data processing procedures is missing. It is very likely that the published measurement uncertainties are incomplete and the overall uncertainties thus underestimated.

       In this paper we use a Monte Carlo simulation (JCGM-101, 2008) in order to get a better estimate of uncertainties in the major ozone absorption cross-section data. This method will allow us to combine the uncertainties from the laboratory measurements with those from the temperature parameterisation that is widely used to interpolate between the laboratory

40  temperatures. The temperature dependence in the UV ozone absorption cross section is usually expressed as a quadratic polynomial (Bass-Paur parameterisation). Having only four to five temperatures available as in the case of BDM and BP a quadratic polynomial will at many wavelengths overfit the data (perfect matches). However, in MC simulation that includes uncertainties from the laboratory measurements more realistic contributions from uncertainties in the temperature parameterisation to the overall uncertainty will be obtained.

[revised manuscript text omitted]

relative standard measurement uncertainty (see Section 3.1). We later assume an overall systematic uncertainty of about 2% (systematic) which is somewhat higher than the 1.3% estimated for DBM and SG (see next two subsections).

**Table 2: Absolute uncertainties reported by Bass and Paur (1985) and Paur and Bass (1985). Light gray highlighting stands for type A , dark gray for type B uncertainties.**

| Uncertainty | Values |
|---|---|
| Wavelength uncertainty | 0.025 nm |
| Uncertainty in the transmittance determination | 2 in $10^5$ (arising from counting statistics) |
| Sample temperature stability | better than 1 K |
| Temperature uncertainty | 0.25 K |
| Pressure measurement uncertainty | 1 mbar |
| Absolute scaling | using the value of Hearn at 253.65 nm |

**3.3    Uncertainty budget of BDM**

Ozone absorption cross-sections provided by Brion et al. (1993), Daumont et al. (1992) and Malicet et al. (1995) further extend the wavelength coverage (into the visible) compared to BP. The following information is available on the experimental details for BDM data in the Hartley-Huggins ozone absorption band:

— Wavelength range: 195 – 520345 nm (except at 273 K: 300–520345 nm)

— Spectral resolution: 0.01 nm

— Wavelength grid: in steps of 0.01 nm

— Concatenation: 15 nm wide spectral cuts, 5 nm overlap

— Number of spectra averaged: 10

— Temperatures: 218 K, 228 K, 243 K, 273 K, 295 K

— Temperature uncertainty: from 0.05 K @ at 295 K to 0.3 K at @ 218 K

— Light source reference spectra: recorded before and after the ozone spectra

— Absolute scaling: measurements of total pressure

Table 3 provides the original notation of the uncertainty budget of BDM data with Type A / Type B breakdown color-coded in gray shadings. The information on the relative standard measurement uncertainty of the BDM ozone absorption cross-section is wavelength-dependent (see last row of Table 3).

**Table 3: Summary of uncertainties as reported by  Daumont et al. (1992) and Malicet et al. (1995). Light gray highlighting stands for type A, dark gray for type B uncertainties.**

|  |  |
|---|---|
|  |  |
|  |  |
|  |  |
|  |  |
|  |   |
|  |   |
|  |   |
|  |  |

| | Daumont et al., (1992) | Malicet et al., (1995) |
|---|---|---|
| **Quantity** | **Uncertainty** | **Uncertainty** |
| Absorbance (for λ < 335 nm) | 1 % | 1 % |
| Optical path | 0.05 % | 0.05 % |
| Ozone pressure | 0.1 % | 0.1 % |
| Impurities | < 0.1 % | < 0.1 % |
| Temperature | 0.02 % | from 0.02 % at 295 K up to 0.15 % at 218 K |
| Wavelength | < 0.05 % (Hartley band) < 0.8 % (Huggins band) | 0.005-0.015 nm |
| Total (systematic) error | 1.3 % (Hartley band) 1.3 – 2.5 % (Huggins band) | 1.3 – 1.5 % (Hartley band) 1.3 – 3.5 % (Huggins band) |
| Random error (RMS) (for λ< 340 nm) | 0.9 – 2.2 % | 0.3 – 2.0 % |

**3.4 Uncertainty budget of SG**

Ozone absorption cross-sections reported by Serdyuchenko et al. (2014) were obtained for 11 temperatures in a wide spectral range using two spectrometers (Fourier-transform and Echelle-grating spectrometers). Tables 4 and 5 summarize the information on the experimental details and uncertainties for the SG cross-section data.

In the 213-350 nm wavelength region the relative standard measurement uncertainty of the SG ozone absorption cross-section is wavelength-dependent and ranges between 1 – 3 %. The dominating uncertainty source is the statistical repeatability of the spectral measurements , influenced by stability of the light source and detector noise. These two factors have a greater impact when the intensities of the spectra contributing to the OD calculation either differ greatly

(strong absorption, close to saturation) or are very close to each other. This effect is demonstrated in Figure 1, showing the concatenated OD spectrum and relative uncertainties of the corresponding constituent spectral cuts. The latter were calculated according to law of propagation of uncertainty using standard deviations  and mean values of I and $I_0$ spectra, respectively, which determine the optical density OD (see Eq. 3).

Table 6 summarises the uncertainties for all cross-sections datasets discussed here.

**Table 4. Experimental details and statistical uncertainty of OD spectra for different wavelength regions for SG data (Serdyuchenko et al., 2014).**

| Spectral region Region, [nm] | Spectrometer, detector | Resolution [nm] | Calibration | Path, [cm] | Lamp stability* [%] | Optical density |
|---|---|---|---|---|---|---|
| 213 – 310 | Echelle, ICCD | 0.018 | Relative | 5 | D₂, 0.5 | 0.5 – 2 |
| 310 – 335 | FTS, GaP | 0.01 | Absolute | 135 | Xe, 2 | 0.1 – 2 |
| 335 – 350 | FTS, GaP | 0.012 | Relative | 270 | Xe, 1 | 0.1 – 1 |
| 350 – 450 | Echelle, ICCD | 0.02 | Relative | ~2000 | Xe, 1 | 0.05 – 1 |
| 450 – 780 | FTS, Si | 0.02-0.06 | Absolute | 270 | W, 0.2 | 0.05 – 2 |
| 780 -1100 | FTS, Si | 0.12-0.24 | Relative | 270 | W, 0.2 | 0.001 – 0.1 |

\* during the entire measurement

$D_2$, Xe - deuterium and xenon discharge lamps, W - tungsten filament lamp

**Table 5. Summary of absolute and relative measurement uncertainties for SG dataset. Dark gray stands for systematic and light gray highlighting stands for type A, dark gray for type B for random uncertainties.**

| Systematic uncertainty (abs) | (rel) [%] | Random uncertainty (abs) | (rel) [%] |
|---|---|---|---|
| Ozone impurity: | 0.005 | Ozone initial pressure | < 1 |
| oxygen impurity leaks | < 0.1 | Pressure fluctuations (< 0.04 mb) | < 0.08 |
| Pressure sensors (0.02 mb) | 0.04 | Temperature fluctuations (<0.3 K) | < 0.1 |
| Temp. sensors offset (1 K) | 0.3 – 0.5 | Light source stability, relative to optical | 0.2 – 2 |
| Temp. non-uniformity (1 K) | 0.3 – 0.5 | density OD = 1 (depending on spectral | |
| Cell length (0.1-1.0 mm) | 0.04 – 0.07 | region) | |
| **Combined standard relative uncertainty** **(excluding low absorption regions near 380nm and above 800nm)** | | | |
| | 0.4 – 0.7 | | 1 – 2.2 |

 **Table 6. Uncertainties for Hearn, BP, BDM and SG ozone cross-sections.**

| Dataset | Scaling method | random (Statistical) | systematic | Relative standard measurement uncertainty, [%] |
|---|---|---|---|---|
| Hearn (253.65 nm) | Absolute, pure ozone | 1.05 | – | 1.4 |
| BP | Using Hearn | 1 | 2.1 | > 2.1 |
| BDM | Absolute, pure ozone | 0.9 - 2.2 | 1.3 (Hartley) | 2 – 3 |
| | | | 1.3 - 3.5 (Huggins) | 2 – 4 |
| SG | Absolute, pure ozone | 1 – 2.2 | 0.4 - 1.7 | 1.1 – 3 |

[Figure]

**Figure 1. Upper panel: concatenated optical density spectrum. Lower panel: relative uncertainty of various OD spectra used for concatenation. Instruments: Echelle/FTS; number of averaged spectra: 2000 (Echelle)/100 (FTS); acquisition time: ~ 30 minutes; light sources: Xe and D₂ lamps. From Serdyuchenko et al. (2014).**

**4    Temperature dependence and uncertainties**

205    In general the ozone cross-sections were determined at selected atmospherically relevant temperatures. BP and BDM data encompass six and five temperatures, respectively, while SG is available at eleven temperatures. The original data can be obtained from *http://satellite.mpic.de/spectral_atlas/cross_sections/Ozone/O3.spc*. Table 7 summarises the available temperatures for all three datasets.

**Table 7. Available temperatures for various ozone absorption cross-section data. Temperatures provided in brackets were not used to determine temperature coefficients since they did not cover the complete wavelength range of 290-340 nm.**

| Ozone absorption cross-section | Temperatures [K] | Wavelength range [nm] |
|---|---|---|
| BP (Paur & Bass, 1985) | 203, 218, (228), 243, 273, 298 | 245 - 343 |
| BDM (Malicet et al., 1995) | 218, 228, 243, (273), 295 | 195 -  520 |
| SG (Serdyuchenko et al., 2014) | 193, 203, 213, 223, 233, 243, 253, 263, 273, 283, 293 | 213 - 1100 |

210

The temperature dependence of the ozone absorption cross-sections is commonly described by the so-called Bass-Paur parameterisation (Paur and Bass, 1985), which is a quadratic polynomial:

$$\sigma_p(\lambda, t) = a_0(\lambda) + a_1(\lambda)\, t + a_2(\lambda)\, t^2. \tag{4}$$

The temperature coefficients $a_0, a_1, a_2$ are determined in a multivariate linear regression using the cross-section data $\sigma(\lambda, t_i)$
215    measured at various temperatures $t_i$. They were calculated in the wavelength range 290-360 nm (BP: up to about 338 nm), which is the spectral range with the highest temperature sensitivity (Huggins ozone band). The temperature t in Eq. 4 is given in degree Celsius (t=T-273.15 K). The uncertainty of the calculated cross-section at a given temperature is then given by

$$\Delta\sigma_p = \sqrt{(\Delta a_0)^2 + (\Delta a_1)^2\, t^2 + (\Delta a_2)^2\, t^4}. \tag{5}$$

The 228 K temperature data of BP has been excluded from the polynomial fit as there is a gap between 295 and 304 nm. Liu
220    et al. (2007) noted a systematic bias in the 273 K BDM data and reported better ozone retrieval fit results if this temperature is excluded. This temperature data also does not provide data below 300 nm. As noted by Orphal and Chance (2003) and Weber et al. (2013), there is a systematic wavelength shift between BP and BDM. Shifting the BP data by +0.029 nm leads to better agreement (to within 0.5%) between BDM and BP (Weber et al., 2013). The SG data wavelength scale agrees to within uncertainties with BDM (Gorshelev et al., 2014).
225    Figures 2 to 4 show the temperature coefficients including the 1-sigma uncertainties (see Eq. 5) for the BP, BDM, and SG data, respectively. As the SG data are somewhat noisy near 300 nm, in the July 2013 version of the SG data a fast

Fourier transform filter was applied in the spectral range 213-317 nm. These figures also show that the BP data appear somewhat noisier than the others and one striking difference between SG and BDM is the apparent bump in the third coefficient (blue line) near 305 nm evident in BDM.

230    Figure 5 shows the uncertainty from the polynomial fit as a function of wavelength for T=193 K and 227 K. BP data uncertainties are getting fairly large above 330 nm reaching nearly 25% for some wavelengths at 220 K and more than 60% at 193 K. The uncertainty of the BDM data ranges between 0 and 2% up to 330 nm, while SG data show a fairly constant uncertainty of about 1% on average at T=227 K. The uncertainties are doubled at the lower temperature. The very low uncertainty for BP and BDM at some wavelength is mainly due to the very low number of available temperatures (4-5) that

235    leads in some cases to overfitting of the data with a quadratic polynomial.

The spectral resolution of the three datasets varies from 0.01 nm (BDM) to 0.05 nm (BP). In order to determine the ozone cross-section at a specific instrument resolution one can either convolve all the various temperature data with the instrument function and then apply the polynomial fit or the coefficient spectra (as shown in Figs. 2-4) are convolved and the polynomial coefficients from the original data are used.

240

[Figure]

**Figure 2. Temperature coefficients and their uncertainty (1-sigma) of the BP data as a function of wavelength λ. For some coefficients and/or selected wavelengths the uncertainties are too small to be visible.**

[Figure]

**Figure 3. Same as Fig. 2, but for BDM data.**

[Figure]

**Figure 4. Same as Fig. 2, but for SG (July 2013 version) data.**

245

[Figure]

**Figure 5. Panels a and b: 1-sigma uncertainty of BP, BDM, and SG ozone cross-sections at T=193 K and 227 K, respectively, from the polynomial temperature fit (Eq. 5). Note the change in scale of the ordinate axes. Panels c and d: Measured ozone cross-sections (points) and polynomial fit (solid lines) for BP, BDM, and SG at 306 and 319.4 nm, respectively.**

**5    Overall uncertainty: Monte – Carlo simulation**

The uncertainty given in Eq. 5 reflects only the uncertainty from the temperature parameterisation using a polynomial (if we assume that a quadratic dependence in temperature is true), thus excluding the experimental uncertainties as discussed in Section 3. One main motivation to only show the uncertainties arising from the polynomial fit is to demonstrate that with only few temperatures available for some of the datasets (BP, DBM) the uncertainty in the temperature dependence is strongly underestimated due to overfitting. In order to estimate the overall uncertainty including uncertainties from

250

measurements (random and systematic), wavelength registration, and the temperature parameterisation, an extensive Monte-Carlo simulation (JCGM-101, 2008) was carried out.

Table 8 summarises the uncertainties simulated. The numbers are mainly based upon the uncertainty as reported in Table 6. It was assumed that the probability density function (PDF) is Gaussian for all uncertainties.

**Table 8. Uncertainties accounted for in the Monte-Carlo simulation assuming a Gaussian PDF.**

| Uncertainty type (290 nm – 370 nm) | Value (1-sigma) |
|---|---|
| Measurement at each T (random) | 1% |
| Measurement at each T (systematic) | 1.3%* |
| Wavelength registration at each T (random) | 0.005 nm |
| Wavelength registration at each T (systematic) | 0 nm** |
| Temperature T (random) | 0.5 K |
| Temperature T (systematic) | 1 K |
| Polynomial in T | combined resampling residuals and wild boot strap (normal distributed) |

* 2% for BP ozone cross-sections

** it is assumed that wavelength shifts can be corrected in ozone retrievals (e.g. Coldewey-Egbers et al., 2015)

The values used here are the minimum uncertainties as summarised in(see Table 6), however, it should be noted that the (random) measurement uncertainties varies with wavelengths, but this is neglected here. Random uUncertainties for all parameters in Table 8 are simulated are drawn from a Gaussian random generator to perturb by randomizing for each the cross section data at each available temperature. data in a given simulation, Random uncertainties means that for each temperature available a new set of random perturbations were calculated, while systematic uncertaintieserrors means an uncertaintiesy drawn from the random generator being were applied to all the temperature data simultaneously. A total of 10,000 perturbed datasets of cross section data were then generated and each fitted by a quadratic polynomial in temperature. The 1σ distributions from the sample polynomials provided then the overall 1σ uncertainty as function of temperature. MC simulations were repeated for all wavelengths between 290 and 370 nm in steps of 0.01 nm. Cross-correlations between adjacent wavelengths, which are difficult to estimate, were neglected.

In order to estimate the effect of the measurement errors on uncertainties from the T-polynomial a combination of the resampling residual and wild boot strap method was applied (Wu, 1986). The residuals from the polynomial fit areis given by

$$\epsilon_i = \sigma(\lambda, t_i) - \sigma_p(\lambda, t_i),$$ (6)

where $\sigma_p(\lambda, t_i)$ is the fitted polynomial (see Eq. 4) and $t_i$ are the temperatures for which the experimental cross section data are available. In the Monte Carlo simulation the residuals areis distributed randomly to different temperatures as follows

$$\sigma'(\lambda, t_i) = \sigma(\lambda, t_i) + \zeta\epsilon_j$$ (7)

in order to perturb the cross-section data.  ζ is a normal distributed random number with mean 0 and variance of 1. The normally distributed random number generator used in the MC simulation is based upon the Box-Muller transform (Box and Muller 1958). The total sample size selected  $10^4$  provide a reasonable compromise between computation time and precision of the simulation. For each wavelength between 290 and 370 nm (BP: ~339 nm) in steps of 0.01 nm the Monte-Carlo simulation was carried out.

[Figure]

[Figure]

**Figure 6. Modelled uncertainties of the three major ozone absorption cross-sections (BP, SG, and BDM) at 319.4 nm based upon MC simulations. Red crosses are the measured data including temperature and measurement uncertainties (here expressed as the square sum of random and systematic errors). The green curve is the fitted polynomial and the black curves show the modelled ±2-sigma uncertainty.**

Figures 6 and 7 show the results from the MC simulation of uncertainties for the three major ozone cross-section data at 319.4 nm and 306 nm, respectively, as an example. These plots are corresponding to the data shown in panels **c** and **d** of Figure 5. The uncertainties are very similar for BDM and SG, except for the lowest temperatures (T < 215 K) where BDM uncertainties increase due to the extrapolation of the fitted polynomial. The larger systematic measurement uncertainty of the BP data (2%), due to the uncertainties related to the Hearn value at the mercury line used for scaling the BP data (see Section 3.1), leads to larger overall uncertainties in the BP data. The uncertainties from the laboratory measurements are

indicated in Figs. 6 and 7 by the red crosses which show the overall uncertainty from the laboratory measurements (systematic and random, vertical bar) as well as the temperature uncertainty (horizontal bar). The overall uncertainties from the MC simulation are generally in good agreement with measurement uncertainty estimates, however, Figure 6a (BP at 319.4 nm) clearly indicates that the overall uncertainty is here larger than the uncertainties from the measurements and this is due to the outlier at 203 K which increases largely the contribution from the uncertainty of the polynomial.

Figure 8 shows the uncertainties as a function of wavelengths for selected temperatures. At 227 K the overall uncertainties are about 1.5% for both BDM and SG, while BP uncertainties are about 2.1% (1-sigma). Above 330 nm the uncertainties increase for all datasets. At very low temperatures, e.g. 193 K, the BDM uncertainties increase to about 4% (1-sigma), while SG remains at 2% (1-sigma). BP uncertainties are about 2.5% (1-sigma) and are also lower than BDM. Similar to 227 K, the uncertainties increase at the longest wavelengths. At temperatures above 215 K, the uncertainties of BDM and SG are very similar, at lower temperatures the BDM uncertainties significantly increases due to the lack of very low temperature measurements.

[Figure]

[Figure]

**Figure 7. Same as Fig. 6, but at 306 nm.**

[Figure]

**6    Summary and conclusion**

Realistic and comparable uncertainty budgets were derived from three major ozone absorption cross-section datasets that are used in various remote sensing applications. First a review of the published literature on the uncertainty of the BP, BDM, and SG datasets was given. The uncertainties of these three datasets are summarized in Table 6 and are now directly comparable between the various datasets.  For remote sensing application, in particular in the Huggins ozone band, the temperature dependence of the ozone cross-sections have to be accounted for and this is typically done using a quadratic polynomial as a function of temperature. Using the updated uncertainty estimates from Table 6 and a residual boot strap method for estimating the uncertainties from the temperature polynomial, a Monte Carlo simulation was carried out. However, one should note that due to lack of information from the peer-review literature  the  wavelength dependence of the uncertainties (see Table 6)  was neglected in our simulations.

[Figure]

**Figure 8. Uncertainty (1-sigma) of BP, BDM, and SG ozone cross-sections at T=193 K (left) and 227 K (right), respectively, from the MC simulation. Note the change in scale of the ordinate axes.**

In the Huggins band the overall uncertainty of the temperature dependent ozone cross-section is about 1.5% (1-sigma) for BDM and SG and 2.1% (1-sigma) for BP up to about 330 nm. At temperatures below about 215 K the uncertainty in the BDM data increase more strongly than for the others, as the lowest measured temperature for BDM is 218 K and the extrapolation of the polynomial leads to larger uncertainties. Above 330 nm the uncertainties increase significantly for all datasets. Ozone retrievals exploiting the UV spectral range usually focus on wavelengths below 335 nm.

In recent years a seemingly larger proportion of publications are adhering to the guidelines of uncertainty reporting as recommended by JCGM-100 (2008). However, there is still either a lack of consistency in the approaches used or uncertainty budgets are not detailed enough. It is especially challenging to re-evaluate uncertainty budgets of older published datasets. In this paper we attempted to provide a more realistic uncertainty budget that may be useful when trying to establish the contribution from ozone absorption cross-sections to the overall uncertainty of retrieved ozone. This work is part of a project on traceability of total ozone measurements (ATMOZ), where we plan several applications of our results in simulated retrievals as well as the general impact on satellite and ground retrievals of total ozone. This will be subject of other publications.

The temperature coefficients for the three major cross-section data with uncertainty estimates from the polynomial fit alone as well as from the MC simulations are available at *http://www.iup.uni-bremen.de/~weber/ATMOZ.*

**335**  **Acknowledgment**

This work was supported in part by a grant from EMRP within the ENV59-ATMOZ ('Traceability for Atmospheric Total Column Ozone') Joint Research Programme (JRP), University of Bremen, and the State Bremen. The EMRP is jointly funded by the EMRP participating countries within EURAMET and the European Union.

**340**  **References**

ACSO: WMO/GAW-IO3C-IGACO-O3 Activity "Absorption Cross Sections of Ozone (ACSO)", http://igaco-o3.fmi.fi/ACSO/, last accessed: November 2015, 2010.

Bass, A. M. and Paur, R. J.: The ultraviolet cross-sections of ozone: I. the measurements, in: Atmospheric Ozone, edited by: Zerefos, C. S. and Ghazi, A., Proc. Quadrennial Ozone Symposium, Halkidiki, Greece, 1984, Reidel, D., Dordrecht, 606–**345**  610, 1985.

Box, G. E. P. and Muller, M. E.: A note on the generation of random normal deviates. Ann. Math. Stat. 29, 610-611, 1958.

Brion, J., Chakir, A., Daumont, D., Malicet, J. and Parisse, C.: High-resolution laboratory absorption cross section of O3. Temperature effect, Chem. Phys. Lett., 213, 610–612, doi:10.1016/0009-2614(93)89169-I, 1993.

Chehade, W., Gür, B., Spietz, P., Gorshelev, V., Serdyuchenko, A., Burrows, J. P., and Weber, M.: Temperature dependent **350**  ozone absorption cross section spectra measured with the GOME-2 FM3 spectrometer and first application in satellite retrievals, Atmos. Meas. Tech., 6, 1623-1632, doi:10.5194/amt-6-1623-2013, 2013.

Chehade, W., Gorshelev, V., Serdyuchenko, A., Burrows, J. P., and Weber, M.: Revised temperature-dependent ozone absorption cross-section spectra (Bogumil et al.) measured with the SCIAMACHY satellite spectrometer, Atmos. Meas. Tech., 6, 3055-3065, doi:10.5194/amt-6-3055-2013, 2013.

**355**  Coldewey-Egbers, M., Weber, M., Lamsal, L. N., de Beek, R., Buchwitz, M., Burrows, J. P.: Total ozone retrieval from GOME UV spectral data using the weighting function DOAS approach, Atmos. Chem. Phys. 5, 5015-5025, 2005.

Daumont, D., Brion, J., Charbonnier, J. and Malicet, J.: Ozone UV spectroscopy I: Absorption cross-sections at room temperature, J. Atmos. Chem., 15, 145–155, doi:10.1007/BF00053756, 1992.

Gorshelev, V., Serdyuchenko, A., Weber, M., and Burrows, J. P.: High spectral resolution ozone absorption cross-sections: **360**  Part I. Measurements, data analysis and comparison around 293K, Atmos. Meas. Tech., 7, 609-624, doi:10.5194/amt-7-609-2014, 2014.

JCGM-100: Evaluation of measurement data – Guide to the expression of uncertainty in measurement (JCGM-100), http://www.bipm.org/utils/common/ documents/jcgm/JCGM_100_2008_E.pdf, last accessed: November 2015, 2008

JCGM-101: Evaluation of measurement data - Supplement 1 to the "Guide to the expression of uncertainty in measurement"
365    - Propagation of distributions using a Monte Carlo method (JCGM-101), http://www.bipm.org/ utils/common/documents/jcgm/JCGM_101_2008_E.pdf, last accessed: October 2015, 2008.

Hearn, A. G.: The absorption of ozone in the ultra-violet and visible regions of the spectrum, Proc. Phys. Soc., 78, 932–940, doi:10.1088/0370-1328/78/5/340, 1961.

Liu, X., Chance, K., Sioris, C. E. and Kurosu, T. P.: Impact of using different ozone cross sections on ozone profile
370    retrievals from Global Ozone Monitoring Experiment (GOME) ultraviolet measurements, Atmos. Chem. Phys., 7, 3571–3578, doi:10.5194/acp-7-3571-2007, 2007.

Malicet, J., Daumont, D., Charbonnier, J., Parisse, C., Chakir, A. and Brion, J.: Ozone UV spectroscopy. II. Absorption cross-sections and temperature dependence, J. Atmos. Chem., 21, 263–273, doi:10.1007/BF00696758, 1995.

Park, S. K. and Miller, K. W.: Random number generators: good ones are hard to find, Commun. ACM, 31, 1192–1201,
375    doi:10.1145/63039.63042, 1988.

Paur, R. J. and Bass, A. M.: The ultraviolet cross-sections of ozone: II. Results and temperature dependence, in: Atmospheric Ozone, edited by: Zerefos, C. S. and Ghazi A., Proc. Quadrennial Ozone Symposium, Halkidiki, Greece, 1984, Reidel, D., Dordrecht, 611–615, 1985.

Redondas, A., Evans, R., Stuebi, R., Köhler, U., and Weber, M.: Evaluation of the use of five laboratory determined ozone
380    absorption cross sections in brewer and dobson retrieval algorithms, Atmos. Chem. Phys., 14, 1635-1648, doi:10.5194/acp-14-1635-2014, 2014.

Serdyuchenko, A., Gorshelev, V., Weber, M., and Burrows, J. P.: New broadband high-resolution ozone absorption cross-sections, Spectroscopy Europe, 23, 14-17, 2011.

Serdyuchenko, A., Gorshelev, V., Weber, M., Chehade, W., and Burrows, J. P.: High spectral resolution ozone absorption
385    cross-sections – Part 2: Temperature dependence, Atmos. Meas. Tech., 7, 625-636, doi:10.5194/amt-7-625-2014, 2014.

Sofen, E. D., Evans, M. J., and Lewis, A. C.: Updated ozone absorption cross section will reduce air quality compliance, Atmos. Chem. Phys., 15, 13627-13632, doi:10.5194/acp-15-13627-2015, 2015.

Viallon, J., Lee, S., Moussay, P., Tworek, K., Petersen, M., and Wielgosz, R. I.: Accurate measurements of ozone absorption cross-sections in the Hartley band, Atmos. Meas. Tech., 8, 1245-1257, doi:10.5194/amt-8-1245-2015, 2015.

390    Weber, M., Chehade, W., Gorshelev, V., Serdyuchenko, A.,  Spietz, P.: Impact of ozone cross-section choice on WFDOAS total ozone retrieval applied to GOME, SCIAMACHY, and GOME-2 (1995-present), Technical Note Issue 2 with updates from    November    2013,    a    contribution    to    IGACO-O3/ACSO,    2013,    http://www.iup.uni-bremen.de/UVSAT_material/technotes/weber_acso_201311.pdf, last accessed: November 2015, 2013.

WMO-GAW: Absorption Cross-Sections of Ozone (ACSO) Status Report June 2015, WMO-GAW Report 218, World
395    Meteorological                        Organization,                        Geneva,                        Switzerland, http://www.wmo.int/pages/prog/arep/gaw/documents/FINAL_GAW_218.pdf, 2015.

Wu, C. F. J.: Jackknife, bootstrap and other resampling methods in regression analysis (with discussions), Ann. Stat. 14, 1261–1350, doi:10.1214/aos/1176350142, 1986.